# Evolution of Antimicrobial Susceptibility to Penicillin in Invasive Strains of *Streptococcus pneumoniae* during 2007–2021 in Madrid, Spain

**DOI:** 10.3390/antibiotics12020289

**Published:** 2023-02-01

**Authors:** Sara de Miguel, Marta Pérez-Abeledo, Belén Ramos, Luis García, Araceli Arce, Rodrigo Martínez-Arce, Jose Yuste, Juan Carlos Sanz

**Affiliations:** 1Epidemiology Department, Directorate General of Public Health, Regional Ministry of Health of Madrid, 28002 Madrid, Spain; 2Department of Preventive Medicine, University Hospital 12 de Octubre, 28041 Madrid, Spain; 3CIBER of Respiratory Diseases (CIBERES), 28029 Madrid, Spain; 4Departamento de Epidemiología y Salud Pública, Epidemiología de las Enfermedades Infecciosas, Universidad de Alcalá, Alcalá de Henares, 28801 Madrid, Spain; 5Clinical Microbiology Unit, Public Health Regional Laboratory of the Community of Madrid, Directorate General of Public Health, Regional Ministry of Health of Madrid, 28055 Madrid, Spain; 6Spanish Pneumococcal Reference Laboratory, National Center for Microbiology, Instituto de Salud Carlos III, 28222 Madrid, Spain; 7CIBER of Epidemiology and Public Health (CIBERESP), 28029 Madrid, Spain

**Keywords:** *Streptococcus pneumoniae*, serotypes, antimicrobial susceptibility, resistance, penicillin

## Abstract

The use of pneumococcal conjugate vaccines has affected the epidemiology and distribution of *Streptococcus pneumoniae* serotypes causing Invasive Pneumococcal Disease (IPD). The aim of this study was to analyze the evolution of the phenotypical profiles of antimicrobial susceptibility to penicillin (PEN) in all IPD strains isolated in Madrid, Spain, during 2007–2021. In total, 7133 invasive clinical isolates were characterized between 2007 and 2021. Levels of PENR and PNSSDR were 2.0% and 24.2%, respectively. In addition, 94.4% of all the PENR belonged to four serotypes, including 11A (33.6%), 19A (30.8%), 14 (20.3%) and 9V (9.8%). All the strains of serotype 11A, which is a non-PCV13 serotype, were detected after the year 2011. Serotypes 6C, 15A, 23B, 24F, 35B, 19F, 16F, 6B, 23F, 24B, 24A, 15F and a limited number of strains of serogroups 16 and 24 (non-typed at serotype level) were associated with PNSSDR (*p* < 0.05). PNSSDR strains of non-PCV13 serotypes 11A, 24F, 23B, 24B, 23A and 16F were more frequent from 2014 to 2021. The changes in *S. pneumoniae* serotype distribution associated with the use of conjugate vaccines had caused in our region the emergence of non-PCV13 pneumococcal strains with different PENR or PNSSDR patterns. The emergence of serotype 11A resistant to penicillin as the most important non-PCV13 serotype is a worrisome event with marked relevance from the clinical and epidemiological perspective.

## 1. Introduction

Vaccination with the 23-valent pneumococcal polysaccharide vaccine (PPV23) was recommended in Spain in 2001 for individuals aged more than two years old who were at high risk of pneumococcal disease [1]. In 2005, some regions, such as the Autonomous Community of Madrid, extended its use to adults over 59 years old. In addition to PPV23, pneumococcal conjugate vaccines (PCVs) have been included in the Madrid region [2,3,4]. First, the 7-valent pneumococcal conjugate vaccine (PCV7) that included serotypes 4, 6B, 9V, 14, 18C, 19F and 23F was introduced in 2001 in the private paediatric market. In November of 2006, Madrid included this vaccine in the children’s vaccination program [2]. The use of PCV7 affected the distribution of serotypes in the Spanish population, with changes in the penicillin susceptibility patterns of *Streptococcus pneumoniae* [5,6]. Thus, the non-susceptibility to penicillin level decreased from more than a half at the beginning of this century to near one third at the end of the first decade [6]. After the use of the PCV7, the rise in the penicillin non-susceptible serotype 19A (not covered by this vaccine) became very prevalent [6]. In 2010, PCV7 was replaced by the 13-valent pneumococcal conjugate vaccine (PCV13), containing serotypes covered by PCV7 plus 1, 3, 5, 6A, 7F and 19A; it was introduced for the paediatric population. In 2012, PCV13 was removed from the funded immunization program in Madrid with private administration according to the individual recommendations of paediatricians and this vaccine was finally implemented in the Spanish national immunization childhood calendar in 2016 [2,3,4]. Its use in children has affected the epidemiology of the invasive pneumococcal disease (IPD), reducing the incidence in children but also in adults due to its herd protection effect. The decrease in PCV13 serotypes since 2010 was accompanied by a decrease in penicillin non-susceptibility within PCV13 isolates [5]. This reduction was mainly due to the fall of the serotype 19A incidence [7]. However, the reduction in covered PCV13 serotypes after the use of this vaccine was followed by the emergence of non-PCV13 serotypes [4,5,8,9]. Antimicrobial resistance has been proposed as one of the top ten public health threats by the World Health Organization. To reinforce the discovery of new antibiotics, artificial intelligence strategies have been proposed to identify new antibiotics and even to predict the evolution of vaccine preventable diseases [10,11,12]. In this context of high rates of antibiotic resistance and serotype replacement by non-vaccine serotypes, the aim of this study was to analyze the evolution of the phenotypical profiles of antimicrobial susceptibility to penicillin in IPD isolates from Madrid, Spain, during the period 2007–2021.

## 2. Results 

The levels of PENR and PNSSDR for all the IPD strains studied from 2007 to 2021 (n = 7133) were 2.0% and 24.2%, respectively. The proportion of PENR strains was higher in 2014 (5.6%) and 2015 (3.8%) and lower in 2008 (0.8%), 2009 (0.7%) and 2011 (0.2%) (Table 1). The proportion of PNSSDR strains was higher in 2013 (32.6%) and 2021 (30.4%) and lower in 2017 (17.2%) and 2019 (19.7%) (Table 1). The MIC50 ranged from 0.016 mg/L in 2009 to 0.032 mg/L in 2013–2015 (Table 1). The MIC90 was above the PNSSDR breakpoint and oscillated from 0.38 mg/L in 2017, 2019 and 2020 to 2 mg/L in 2014 (Table 1). 

Serotypes 11A, 14, 9V and 19A were associated with PENR (resistance 22.6%, 18.6%, 14.4% and 7.8%, respectively) (*p* < 0.05) (Table 2). In these four serotypes, the MIC50 corresponded to PNSSDR of high level. In serotypes 11A, 14 and 9V, the MIC90 was 3 mg/L (above the PENR CLSI and EUCAST breakpoints).

Serotypes 6C, 15A, 23B, 24F, 35B, 19F, 16F, 6B, 23F, 24B, 24A, 15F and non-fully typed strains of serogroups 16 and 24 were associated with PNSSDR (*p* < 0.05) (Table 2). For the majority of strains (except 19F and 16F), the MIC50 was above the PNSSDR breakpoint (and corresponded to low-level PNSSDR criteria). In serotypes 15A, 24F, 35B, 19F, 6B, 23F, 24B, 24A, 15F and non-fully typed strains of serogroup 24, the MIC90 corresponded to high level PNSSDR. Temporal distribution of PENR IPD strains according to serotype was also evaluated. Thus, 94.4% of all the PENR belonged to serotypes 11A (33.6%), 19A (30.8%), 14 (20.3%) and 9V (9.8%) (Table 3). We also observed trend variations for some of these serotypes with reduced susceptibility. PENR strains of serotype 11A were firstly detected in the early PCV13 period (2010–2012), after the year 2011, whereas PENR strains of serotype 19A were found every year except 2012 and 2021 (Figure 1). The proportion of PENR strains of serotype 14 decreased from the beginning of the PCV13 use in 2010 when compared to 2007. PENR strains of serotype 14 were less frequent during the last years (only one strain during the late PCV13 period (2017–2019) and none during the COVID-19 period (2020–2021)) (Figure 1). PENR isolates of serotype 9V were more frequent from the middle of the PCV13 period (after 2013; 10 out of 14 strains) (Figure 1). The only two PENR isolates of serotype 6B (contained in PCV7 and PCV13) appeared in the late PCV7 period during the years 2007 and 2008. For serotype 8 (non-PCV13 serotype), the only PENR strain of our study was isolated in 2007. Finally, we did not observe significant trend changes across time for other serotypes.

In this study, we also evaluated the distribution across time of 21 serotypes/serogroups that included 94.8% of all the PNSSDR strains (Table 4). The majority of PNSSDR strains belonging to PCV13 were detected during the late PCV7 and early PCV13 periods (2007–2011) and were associated with serotypes 19A, 14, 19F, 6B and 23F. In contrast, PNSSDR strains of non-PCVs serotypes (11A, 24F, 23B, 24B, 23A, 16F and non-fully typed strains of serogroup 16) were more frequent in the last years from the middle and late PCV13 periods (between 2014 and 2019) (Table 4). In addition, PNSSDR strains of serotypes 6C, 15A, 9V and 35B showed a uniform pattern of detection for all the study series. Non-fully typed PNSSDR strains of serogroup 24 were most prevalent during the first years (Table 4).

## 3. Discussion

Microbiological surveillance studies deciphering circulating serotypes and resistance profiles are critical for understanding local epidemiology of IPD, for assessing the impact of current and future vaccines and for monitoring antimicrobial susceptibility. However, different breakpoints to penicillin in *S. pneumoniae* depending on the guidelines (EUCAST or CLSI), variations in national immunization calendars by country and the implementation of different surveillance systems make the comparison of results difficult [13,14]. In the year 2008, CLSI changed the penicillin cut-off that was largely used until 2007 [15,16]. In 2020, EUCAST introduced a change in the intermediate criteria considering the strains in this category as susceptible with increased exposure, assuming that there is a high probability of therapeutic success by increasing the antimicrobial concentration [17]. This change has been maintained in the 11th EUCAST 2021 version [18]. The main reason for this recommendation is to detect treatable infection rather than the identification of resistance mechanisms [14]. The use of the old definition of “intermediate” crafted by EUCAST 2002–2018 [19] had proved to be difficult in clinical practice and EUCAST now categorises as “susceptible increased exposure” when there is a high likelihood of therapeutic success, because exposure to the agent is increased by adjusting the dosing regimen or by its concentration at the site of infection. However, previously, surveillance systems considered the categories intermediate and resistant under a wide definition of non-susceptibility [20]. In this study, in order to maintain traceability with the scientific literature, it has been chosen to consider the categories of PENR and PNSSDR (this last corresponded to the term “non-susceptible” usually applied). In other studies using the same antimicrobial methodology, resistant and intermediate isolates were all referred to as non-susceptible [21,22]. Although E-test has been widely used to perform antimicrobial susceptibility testing of *S. pneumoniae*, this method is not considered by CLSI or EUCAST in their guidelines, and, therefore, it may represent a limitation of the current study.

In terms of pneumococcal epidemiology and the contribution of vaccines to ameliorate the antimicrobial resistance problem, due to its herd protection, the use of pneumococcal conjugate vaccines has led to a decrease in the incidence of vaccine serotypes associated with the decline in antibiotic resistance [23,24]. However, after the introduction of the pneumococcal conjugate vaccines (PCV7 and, later, PCV13), a rise in non-vaccine serotypes displaying antibiotic-resistant has been identified [24,25]). A clear example of this phenomenon is represented by serotype 19A (included in PCV13 but not in PCV7). This was one of the most prevalent PENR serotypes after the use of PCV7 [26]. The variation in the incidence and antibiotic resistance of this serotype has been very evident with the introduction of each vaccine in the childhood immunization schedule [4,26,27]. This pattern is clearly reflected in our study, performed using clinical isolates from Madrid, confirming a decrease in the number of IPD cases of serotype 19A and the association of antibiotic resistance after the introduction of PCV13. 

Epidemiological analysis confirmed that changes in the distribution of *S. pneumoniae* serotypes associated with the use of conjugate vaccines had caused in our region the selection of strains belonging to other serotypes with different PENR or PNSSDR patterns. Others authors have noted that susceptibility patterns are serotype-specific [28]. In any case, mutations in the genes coding the penicillin-binding proteins (PBPs) have been recognized in *S. pneumoniae* as the major resistance mechanism for β-lactam antibiotics [29], indicating that the emergence of resistant strains can be spread by clonal propagation [30,31]. Thus, the main limitation of the present study is that the analysis was performed at the phenotype level with no information about the molecular resistance mechanisms or the genotypes involved.

In addition, the association of some serotypes such as 11A, 14, 24F and 23B with non-susceptibility to penicillin has been previously described [5,25,32]. In this sense, the increase in beta-lactam resistance among serotype 11A was linked to a clonal shift in this serotype [23,25,33]. In Spain, serotype 11A strains isolated in 2010–2011 already showed a penicillin MIC90 coinciding with the resistant EUCAST breakpoint of 2mg/L [6]. This serotype is currently among the most prevalent causing IPD in our country [4]. Thus, the emergence of penicillin-resistant strains of serotype 11A is concerning from a pathogenesis perspective [34]. The invasive disease potential of this serotype is highly related to the rise of genotype ST6521^11A^ that has spread across Europe in the last years, becoming one of the most prevalent within serotype 11A [35]. This genotype of serotype 11A is associated with high levels of antibiotic resistance, shows a greater ability to form biofilms and avoids very efficiently the host immune response [34]. In a recent study conducted by our group, this prevalent serotype was detected with the second highest fatality rate [36] and has shown an increase in penicillin resistance in the last years. Indeed, the pandemic of SARS-CoV-2 has increased the resistance levels of circulating strains of serotype 11A in Spain, with an increase in MIC90 to penicillin from 2mg/L during 2016–2019 to 4mg/L in 2020 [26]. Another alarming non-PCV13 serotype that has emerged within the PNSSDR strains is 24F. This serotype displays resistance to penicillin and its prevalence in the paediatric and adult population is increasing in some countries, including Spain [4,26].

In the last years, there has been a decrease in the incidence of many of the resistant strains but also in the incidence of IPD due to the COVID-19 pandemic [37]. The surveillance of the behaviour of serotypes and their resistance to antibiotics will be crucial for the development of new vaccines and the implementation of vaccination schedules that can help to prevent IPD. The use of vaccines in national immunization schedules is a cost-effective measure to decrease antibiotic resistance [38,39], and, probably, the inclusion of serotypes 11A and 24F in future vaccines for more of the population could be crucial for preventing penicillin resistance and non-susceptibility [40,41].

## 4. Materials and Methods

Invasive pneumococcal strains from IPD cases (one for every episode) isolated from 2007 to 2021 were sent from the Microbiological Laboratory Services of Public and Private Hospitals located in the Madrid Region (Spain) to the Madrid Public Health Regional Laboratory for serotyping and antimicrobial susceptibility. In this study, 7,133 clinical isolates from IPD were characterized. Identification of the capsular serotypes was carried out by the Pneumotest-Latex (Statens Serum Institut, Copenhagen, Denmark) and by Quellung reaction using commercial antisera (Statens Serum Institut, Copenhagen, Denmark). 

To perform the antimicrobial susceptibility testing of *S. pneumoniae* by the E-test method, commercial strips (Benzylpenicillin ETEST^®^ strips; bioMérieux España S.A) with a concentration rank of 0.002–32mg/L were used. The inoculum was adjusted to a bacterial concentration of 0.5 McFarland standard (or 1 McFarland standard if mucoid strain) and the S. *pneumoniae* ATCC 49619 was employed as the reference strain. The strips were applied to the surface of inoculated Mueller–Hinton, supplemented with 5% of sheep blood. Agar plates were incubated at 35 ± 2 °C in a 5%CO2 atmosphere for 20 to 24h. MIC values were obtained from the scale at the intersection point between the complete inhibition ellipse edge and the strip. According to the CLSI breakpoints (PEN non-susceptibility for the oral treatment of non-meningitis syndromes) [42] and the EUCAST breakpoints (Appendix A) [17,18], pneumococcal strains showing MIC > 2mg/L and >0.06mg/L were categorized, respectively, as PEN resistant (PENR) and PEN non-susceptible at standard dosing regimen (PNSSDR). *S. pneumoniae* strains with MICs 0.094–0.5mg/L were considered PNSSDR of a low level. Pneumococcal isolates with MIC > 0.5–2mg/L were considered PNSSDR of a high level. To analyze the evolution of serotypes causing IPD and the pattern of penicillin susceptibility during the period 2007 to 2021, the Odds Ratio (OR) with its correspondent 95% confidence intervals (CI95) were calculated. The statistical significance was set at *p* < 0.05. Statistical analyses were performed using STATA v16. In order to relate the use of the conjugate vaccines across time, the entire temporal series was divided into the late PCV7 period (years 2007–2009), early PCV13 period (years 2010–2012), middle PCV13 period (years 2013–2016), late PCV13 period (years 2017–2019) and COVID-19 pandemic period (years 2020–2021).

## Figures and Tables

**Figure 1 antibiotics-12-00289-f001:**
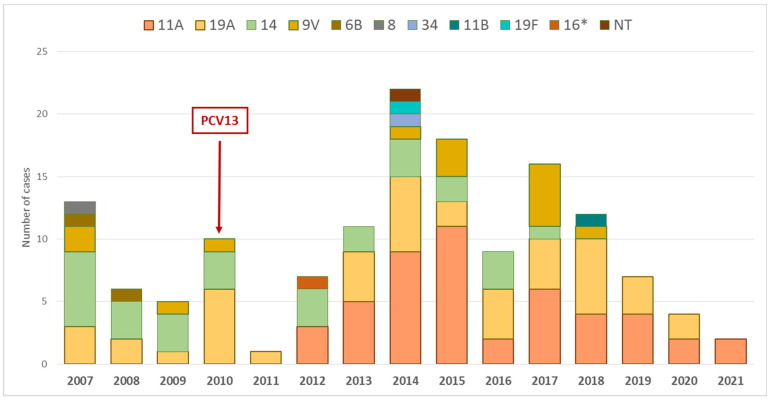
Evolution of the PENR serotypes across time. (Region of Madrid, 2007–2021.)

**Table 1 antibiotics-12-00289-t001:** Evolution of PENR and/or PNSSDR among *S. pneumoniae* invasive strains (region of Madrid, 2007–2021).

Year	Invasive Strains (n)	Penicillin MIC_50_	Penicillin MIC_90_	PENR (n)	PENR (%)	Odds Ratio of PENR (CI95)	PNSSDR (n)	PNSSDR (%)	Odds Ratio of PNSSDR (CI95)
2007–2021	7133	0.023	0.75	143	2.0	NA	1726	24.2	NA
2007	539	0.023	1	13	2.4	1.2 (0.7–2.2)	136	25.2	1.1 (0.9–1.2)
2008	710	0.023	0.75	6	0.8	0.4 (0.2–0.9)	173	24.4	1 (0.9–1.2)
2009	730	0.016	0.75	5	0.7	0.3 (0.1–0.8)	174	23.8	1 (0.9–1.1)
2010	482	0.023	1.5	10	2.1	1 (0.5–2)	142	29.5	1.3 (1.2–1.5)
2011	466	0.023	1	1	0.2	0.1 (0–0.7)	123	26.4	1.1 (1–1.3)
2012	366	0.023	1	7	1.9	1 (0.4–2)	99	27.0	1.2 (1–1.4)
2013	331	0.032	1.5	11	3.3	1.7 (0.9–3.2)	108	32.6	1.6 (1.3–1.8)
2014	394	0.032	2	22	5.6	3.2 (2–5.2)	106	26.9	1.2 (1–1.4)
2015	468	0.032	0.75	18	3.8	2.1 (1.3–3.5)	125	26.7	1.2 (1–1.3)
2016	504	0.023	0.5	9	1.8	0.9 (0.4–1.7)	108	21.4	0.8 (0.7–1)
2017	548	0.023	0.38	16	2.9	1.5 (0.9–2.6)	94	17.2	0.6 (0.5–0.8)
2018	591	0.023	0.5	12	2.0	1 (0.6–1.8)	119	20.1	0.8 (0.7–0.9)
2019	633	0.023	0.38	7	1.1	0.5 (0.2–1.1)	125	19.7	0.8 (0.6–0.9)
2020	210	0.023	0.38	4	1.9	0.9 (0.3–2.6)	45	21.4	0.9 (0.7–1.1)
2021	161	0.023	0.5	2	1.2	0.6 (0.1–2.5)	49	30.4	1.4 (1.1–1.8)

Penicillin resistant (PENR); penicillin non-susceptibility at standard dosing regimen (PNSSDR); odds ratio (OR) with its correspondent 95% confidence interval (CI95); NA (not applicable).

**Table 2 antibiotics-12-00289-t002:** Invasive serotypes significantly associated with PENR and/or PNSSDR. (Region of Madrid, 2007–2021.).

Serotype	IPD (n)	Penicillin MIC50 (mg/L)	Penicillin MIC90 (mg/L)	PENR (n)	PENR (%)	OR of PENR (CI95)	PNSSDR (n)	PNSSDR (%)	OR of PNSSDR (CI95)
All Serotypes	7133	0.023	0.75	143	2.0	NA	1726	24.2	NA
19A	567	0.75	2	44	7.8	5.5 (3.8–7.9)	442	78.0	14.5 (11.8–17.9)
11A	212	2	3	48	22.6	21 (14.4–30.8)	150	70.8	8.2 (6.1–11.1)
14	156	1.5	3	29	18.6	13.7 (8.8–21.4)	145	92.9	45 (24.3–83.3)
9V	97	1.5	3	14	14.4	9 (5–16.3)	77	79.4	12.6 (7.7–20.6)
6C	194	0.094	0.19	0	0.0	0.0	134	69.1	7.5 (5.5–10.2)
15A	164	0.19	1.5	0	0.0	0.0	99	60.4	5 (3.6–6.9)
23B	157	0.19	0.25	0	0.0	0.0	123	78.3	12.1 (8.3–17.8)
24F	138	0.38	1	0	0.0	0.0	117	84.8	18.6 (11.7–29.8)
35B	117	0.5	1	0	0.0	0.0	73	62.4	5.4 (3.7–7.9)
19F	107	0.064	1	1	0.9	0.5 (0.1–3.3)	52	48.6	3 (2.1–4.4)
16F	110	0.032	0.38	0	0.0	0.0	52	47.3	2.9 (2–4.2)
16 *	44	0.25	0.38	1	2.3	1.1 (0.2–8.3)	31	70.5	7.6 (4–14.5)
6B	32	0.125	1.5	2	6.3	3.3 (0.8–13.9)	26	81.3	13.8 (5.7–33.5)
23F	34	0.38	1.5	0	0.0	0.0	27	79.4	12.3 (5.3–28.2)
24B	27	0.38	0.75	0	0.0	0.0	23	85.2	18.2 (6.3–52.8)
24 *	18	0.38	0.5	0	0.0	0.0	16	88.9	25.3 (5.8–110.1)
24A	7	0.38	0.75	0	0.0	0.0	6	85.7	18.9 (2.3–156.8)
15F	6	0.125	1	0	0.0	0.0	4	66.7	6.3 (1.1–34.3)

* Serogroup (non-typed at serotype level); penicillin resistant (PENR); penicillin non-susceptibility at standard dosing regimen (PNSSDR); odds ratio (OR) with its correspondent 95% confidence interval (CI95); NA (not applicable).

**Table 3 antibiotics-12-00289-t003:** Evolution of PENR *S. pneumoniae* invasive strains (number and %) according to serotype (region of Madrid, 2007–2021).

Serotype	2007	2008	2009	2010	2011	2012	2013	2014	2015	2016	2017	2018	2019	2020	2021	2007–2021
11A	0	0	0	0	0	3 (43%)	5 (45%)	9 (41%)	11 (61%)	2 (22%)	6 (38%)	4 (33%)	4 (57%)	2 (50%)	2 (100%)	48 (34%)
19A	3 (23%)	2 (33%)	1 (20%)	6 (60%)	1 (100%)	0	4 (36%)	6 (27%)	2 (11%)	4 (44%)	4 (25%)	6 (50%)	3 (43%)	2 (50%)	0	44 (31%)
14	6 (46%)	3 (50%)	3 (60%)	3 (30%)	0	3 (43%)	2 (18%)	3 (14%)	2 (11%)	3 (33%)	1 (6%)	0	0	0	0	29 (20%)
9V	2 (15%)	0	1 (20%)	1 (10%)	0	0	0	1 (5%)	3 (17%)	0	5 (31%)	1 (8%)	0	0	0	14 (10%)
6B	1 (8%)	1 (17%)	0	0	0	0	0	0	0	0	0	0	0	0	0	2 (1%)
8	1 (8%)	0	0	0	0	0	0	0	0	0	0	0	0	0	0	1 (1%)
34	0	0	0	0	0	0	0	1 (5%)	0	0	0	0	0	0	0	1 (1%)
11B	0	0	0	0	0	0	0	0	0	0	0	1 (8%)	0	0	0	1 (1%)
19F	0	0	0	0	0	0	0	1 (5%)	0	0	0	0	0	0	0	1 (1%)
16 *	0	0	0	0	0	1 (14%)	0	0	0	0	0	0	0	0	0	1 (1%)
NT	0	0	0	0	0	0	0	1 (5%)	0	0	0	0	0	0	0	1 (1%)
All	13	6	5	10	1	7	11	22	18	9	16	12	7	4	2	143

* Serogroup: non-typed at serotype level. NT: non-typed.

**Table 4 antibiotics-12-00289-t004:** Evolution of PNSSDR *Streptococcus pneumoniae* invasive strains (number and %) according to serotype (serotypes with ≥1% of the total PNSSDR strains).

Serotype	2007	2008	2009	2010	2011	2012	2013	2014	2015	2016	2017	2018	2019	2020	2021
19A	39 (28%)	74 (43%)	87 (50%)	67 (47%)	44 (36%)	26 (26%)	17 (16%)	16 (15%)	13 (10%)	12 (11%)	11 (12%)	15 (13%)	15 (12%)	4 (9%)	2 (4%)
14	27 (20%)	22 (13%)	19 (11%)	10 (7%)	7 (6%)	5 (5%)	6 (6%)	9 (8%)	9 (7%)	6 (6%)	4 (4%)	9 (8%)	8 (6%)	2 (4%)	2 (4%)
11A	0	0	5 (3%)	7 (5%)	9 (7%)	15 (15%)	10 (9%)	14 (13%)	19 (15%)	14 (13%)	10 (11%)	18 (15%)	17 (14%)	3 (7%)	9 (18%)
6C	11 (8%)	7 (4%)	3 (2%)	5 (4%)	16 (13%)	11 (11%)	15 (14%)	12 (11%)	9 (7%)	6 (6%)	4 (4%)	11 (9%)	12 (10%)	5 (11%)	7 (14%)
24F	3 (2%)	10 (6%)	2 (1%)	4 (3%)	5 (4%)	2 (2%)	7 (6%)	7 (7%)	16 (13%)	19 (18%)	14 (15%)	15 (13%)	5 (4%)	6 (13%)	2 (4%)
23B	2 (1%)	2 (1%)	5 (3%)	6 (4%)	7 (6%)	7 (7%)	10 (9%)	11 (10%)	14 (11%)	11 (10%)	12 (13%)	4 (3%)	14 (11%)	5 (11%)	13 (27%)
15A	6 (4%)	4 (2%)	8 (5%)	7 (5%)	4 (3%)	11 (11%)	9 (8%)	7 (7%)	10 (8%)	5 (5%)	10 (11%)	5 (4%)	10 (8%)	0	3 (6%)
9V	13 (9%)	9 (5%)	8 (5%)	7 (5%)	2 (2%)	2 (2%)	3 (3%)	4 (4%)	4 (3%)	4 (4%)	8 (9%)	5 (4%)	7 (6%)	0	1 (2%)
35B	4 (3%)	6 (3%)	8 (5%)	8 (6%)	10 (8%)	3 (3%)	5 (5%)	6 (6%)	6 (5%)	5 (5%)	3 (3%)	4 (3%)	4 (3%)	0	1 (2%)
19F	9 (7%)	13 (8%)	3 (2%)	4 (3%)	3 (2%)	2 (2%)	2 (2%)	6 (6%)	3 (2%)	2 (2%)	0	1 (1%)	0	4 (9%)	0
16 *	0	0	0	0	0	3 (3%)	3 (3%)	1 (1%)	2 (2%)	10 (9%)	3 (3%)	9 (8%)	0	0	0
16F	2 (1%)	1 (1%)	1 (1%)	1 (1%)	1 (1%)	0	5 (5%)	3 (3%)	11 (9%)	2 (2%)	3 (3%)	2 (2%)	13 (10%)	6 (13%)	1 (2%)
6B	6 (4%)	4 (2%)	6 (3%)	1 (1%)	3 (2%)	1 (1%)	1 (1%)	2 (2%)	0	0	0	1 (1%)	0	0	1 (2%)
23F	5 (4%)	7 (4%)	2 (1%)	3 (2%)	1 (1%)	1 (1%)	0	1 (1%)	1 (1%)	0	1 (1%)	1 (1%)	1 (1%)	3 (7%)	0
24B	1 (1%)	1 (1%)	0	0	1 (1%)	0	0	1 (1%)	3 (2%)	0	2 (2%)	4 (3%)	8 (6%)	2 (4%)	0
23A	0	0	1 (1%)	2 (1%)	0	2 (2%)	1 (1%)	0	2 (2%)	3 (3%)	1 (1%)	4 (3%)	1 (1%)	0	0
24 *	2 (1%)	4 (2%)	3 (2%)	2 (1%)	3 (2%)	0	1 (1%)	1 (1%)	0	0	0	0	0	0	0
6A	0	0	3 (2%)	0	1 (1%)	0	0	1 (1%)	1 (1%)	1 (1%)	1 (1%)	1 (1%)	0	1 (2%)	0
7F	1 (1%)	2 (1%)	5 (3%)	0	0	0	0	0	0	0	0	0	0	0	0
8	1 (1%)	0	0	0	1 (1%)	1 (1%)	1 (1%)	0	0	0	0	2 (2%)	0	1 (2%)	0
15B	0	0	0	1 (1%)	2 (2%)	1 (1%)	1 (1%)	0	0	0	0	0	1 (1%)	1 (2%)	1 (2%)

* Serogroup: non-typed at serotype level.

## Data Availability

Not applicable.

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
