# Peer review of "Evolution of Antimicrobial Susceptibility to Penicillin in Invasive Strains of *Streptococcus pneumoniae* during 2007–2021 in Madrid, Spain"

_antibiotics, 2023, doi:10.3390/antibiotics12020289_

Round 1

Reviewer 1 Report

Article describing the annual rates and the serotype distribution of penicillin non-susceptible and resistant S. pneumoniae isolates from Madrid for a period of 15 years.

In general the article is well written and comprehensible.

Major comments

The authors have carried out susceptibility studies using the E-test method. Neither EUCAST nor CLSI describe this technique to perform susceptibility testing. Furthermore, and although E-test has been widely used for antimicrobial susceptibility testing of S. pneumoniae, this technique raised some controversy for determining beta-lactam antibiotics MICs values (1,2). So I recommend the authors to detail more precisely how they performed the E-test: were the inocula adjusted to any bacterial concentration, what commercial strips were used, what strains were used as quality control, …

1.    Tandé D, et al. Evaluation of the E-test for routine testing of the susceptibility of Streptococcus pneumoniae to benzylpenicillin, amoxicillin and cefotaxime. Clin Microbiol Infect. 1997;3:474-479.

2.       Clavier B, et al. Comparison of the results of the Etest and the method for determining minimum inhibitory concentrations in solid media for penicillin G, amoxicillin, and cefotaxime for S. pneumoniae. A multicenter study. Pathol Biol (Paris). 1998;46:369-74. 

As there is no description of the penicillin-susceptible serotypes, the global picture of vaccines efficacy is somehow distorted.

Minor comments

Lines 42-44. Please, add a reference of this recommendation.

Line 57.  “In 2012, PCV13 was removed from the funded immunization program…”. I suggest adding “in Madrid” as the funded immunization program had not been applied in all Autonomous Communities.

Lines 62, 150: Replace “herd immunity” with “herd protection”.

Lines 79 and 92. Please, show the statistical used and its value before the p<0.05.

Table 1. I guess that the Title of 9th column should read PNSSDR (%) instead of PENR (%).

Lines 104-106. I suggest moving the sentence “The proportion of PENR strains of serotype 14 decreased from the beginning of the PCV13 use in 2010 when compared to 2007.” before the sentence in lines 101-103 “PENR strains of serotype 14 were less frequent during the last years (only one strain along late PCV13 period [2017-102 2019] and none during the COVID-19 period [2020-2021]).” to maintain an order in the description or resistance along time for each serotype. Also, both sentences could be merged into one.

Line 114. Figure 1 shows the evolution of PENR isolates, not of PNSSDR isolates.

Lines 134-141. I suggest adding a table (can be as supplemental material) showing the dates and MIC breakpoint criteria changes of CLSI and EUCAST.

Line 142. Have the authors any reference for assessing that the EUCAST definition of ‘intermediate’ had proved “to be difficult in clinical practice”? What difficulties have been shown? Please, explain clearly.

Lines 152-155. “However, after the decrease caused by the introduction of the PCV13, a rise of antibiotic-resistant non-vaccine covered serotypes has been identified (20,21). A clear example of this phenomenon is represented by serotype 19A, (included in PCV13 but not in PCV7) that is one of the most prevalent PENR serotype (22).”.

These two consecutive sentences seems somehow contradictory: rise of non-PCV13 serotypes but high prevalence of PCV13 serotype 19A . Please, rewrite clearly.

Line 172. Replace “descript” with “described” o similar.

Line 179. “…because divert very efficiently the host immune response (30).” The reference mentioned does not exactly say that; they say in their abstract that “A different pattern of evasion of complement immunity and phagocytosis was observed between (serotype 11A) genotypes.”. As in the present work there is not study of genotypes within serotype 11A, this sentence should be better explained.

Line 191. “…can help to prevent the most difficult cases of IPD”.  Please, rewrite clearly the meaning of  “most difficult cases”: most severe, most antibiotic-resistant, …

Line 192. “The use of vaccines in national immunization schedules is a cost-effective measure to decrease antibiotic resistance”. Please, add a reference that demonstrates this assessment.

Lines 212-217. “To analyze the evolution of serotypes causing IPD and the pattern of penicillin susceptibility during the period 2007 to 2021, the Odds Ratio (OR) with its correspondent 95% confidence intervals (CI95) were calculated.”

It is not clear to this reviewer the need of the OR calculation in tables 1 and 2. What do they represent? In general, the odds ratio is used to measure an association between an exposure and an outcome (case-control studies). Other statists could be more appropriate for the comparison of frequencies.

Lines 214-216. “The extended Mantel-Haenszel Chi Square for Trend (Chi Trend) was used to evaluate the evolution along time of serotypes in comparison to the rest of the penicillin resistant or PNSSDR strains.”

Please, replace “penicillin resistant” with the PENR abbreviator used all through the text.

I cannot see any analysis of the evolution along time of serotype in the text. If no results of this analysis are going to be shown, this can be deleted from Material and methods.

Author Response

Dear Reviewer,

It is a pleasure for us to have the opportunity to present our results in this journal. We appreciate and we receive all suggestions with interest to improve our work. We hope that all the answers and changes made are to your liking and we hope that this work will be interesting for publication in this journal.

We present below the anwers to each point but we hace also attached all the answers in the attached document in case it is easier for you to review.

REEVIEW:

Major comments

The authors have carried out susceptibility studies using the E-test method. Neither EUCAST nor CLSI describe this technique to perform susceptibility testing. Furthermore, and although E-test has been widely used for antimicrobial susceptibility testing of S. pneumoniae, this technique raised some controversy for determining beta-lactam antibiotics MICs values (1,2). So I recommend the authors to detail more precisely how they performed the E-test: were the inocula adjusted to any bacterial concentration, what commercial strips were used, what strains were used as quality control, …

To respond these questions, the following sentence has been included in materials and methods:

To perform the antimicrobial susceptibility testing of S. pneumoniae by the E-test method, commercial strips (Benzylpenicillin ETEST® strips; bioMérieux España S.A) with a concentration rank of 0.002-32 mg/L were used. The inoculum was adjusted to a bacterial concentration of 0.5 McFarland standard (or 1 McFarland standard if mucoid strain) and the S. pneumoniae ATCC 49619 was employed as reference strain. The strips were applied to the surface of inoculated Mueller-Hinton, supplemented with 5% of sheep blood. Agar plates were incubated at 35±2 °C in a 5%CO2 atmosphere during 20 to 24 h. MIC values were obtained from the scale at the intersection point between the complete inhibition ellipse edge and the strip.

In the discussion, a new sentence has been introduced:

Although E-test has been widely used to perform antimicrobial susceptibility testing of S. pneumoniae, this method is not considered by CLSI or EUCAST in their guidelines, and therefore, it may represent a limitation for the current study.

As there is no description of the penicillin-susceptible serotypes, the global picture of vaccines efficacy is somehow distorted.

We thank the Reviewer for the suggestion but the main goal of the study was to evaluate the epidemiology of non-susceptible pneumococcal serotypes in a similar way than a recent study at national level (Sempere J et al Lancet Microbe 2022) in order to compare the results from Madrid with those obtained in the entire country. 

Minor comments

Lines 42-44. Please, add a reference of this recommendation.

Ok. Done

Line 57.  “In 2012, PCV13 was removed from the funded immunization program…”. I suggest adding “in Madrid” as the funded immunization program had not been applied in all Autonomous Communities.

Ok. Done

Lines 62, 150: Replace “herd immunity” with “herd protection”.

Ok. Done

Lines 79 and 92. Please, show the statistical used and its value before the p<0.05.

We thank the Reviewer for the observation. The statistical used is specified in the material and methods section and the values ​​obtained are detailed in the corresponding tables, so we would prefer to keep the current structure so as not to complicate the text.

Table 1. I guess that the Title of 9th column should read PNSSDR (%) instead of PENR (%).

We thank the Reviewer for the observation because it is a mistake. We have changes to PNSSDR (%)

Lines 104-106. I suggest moving the sentence “The proportion of PENR strains of serotype 14 decreased from the beginning of the PCV13 use in 2010 when compared to 2007.” before the sentence in lines 101-103 “PENR strains of serotype 14 were less frequent during the last years (only one strain along late PCV13 period [2017-102 2019] and none during the COVID-19 period [2020-2021]).” to maintain an order in the description or resistance along time for each serotype. Also, both sentences could be merged into one.

Ok. Done

Line 114. Figure 1 shows the evolution of PENR isolates, not of PNSSDR isolates.

We appreciate the suggestion of the reviewer. We have modified the sentence according to the suggestion.

Lines 134-141. I suggest adding a table (can be as supplemental material) showing the dates and MIC breakpoint criteria changes of CLSI and EUCAST.

We appreciate the suggestion of the reviewer. We have added a table as supplementary material

Penicillin CMI (µg/ml)

*S

*I

R

CLSI MIC breakpoints criteria (M100 30th Edition year 2020)

CLSI MIC breakpoints (parenteral non-meningitis)

<2

4

>8

CLSI MIC breakpoints (parenteral meningitis)

<0.06

>0.12

CLSI MIC breakpoints (oral penicillin V)

<0.06

0,12- 1

>2

EUCAST MIC breakpoints criteria (Edition 2021)

EUCAST MIC breakpoints (indications other than meningitis)

<0.06

>2

EUCAST MIC breakpoints (for meningitis)

<0.06

>0.06

*S: Susceptible

*I:  Intermediate or Susceptible at increased exposure by adjusting the dosing regimen

Line 142. Have the authors any reference for assessing that the EUCAST definition of ‘intermediate’ had proved “to be difficult in clinical practice”? What difficulties have been shown? Please, explain clearly.

To explain its idea the next phrase has been added:

The use of the old definition of ‘intermediate’ crafted by EUCAST 2002-2018 (15) had proved to be difficult in clinical practice and EUCAST now categorised as “Susceptible, increased exposure* when there is a high likelihood of therapeutic success because exposure to the agent is increased by adjusting the dosing regimen or by its concentration at the site of infection.

Lines 152-155. “However, after the decrease caused by the introduction of the PCV13, a rise of antibiotic-resistant non-vaccine covered serotypes has been identified (20,21). A clear example of this phenomenon is represented by serotype 19A, (included in PCV13 but not in PCV7) that is one of the most prevalent PENR serotype (22).”.

These two consecutive sentences seems somehow contradictory: rise of non-PCV13 serotypes but high prevalence of PCV13 serotype 19A. Please, rewrite clearly.

We have modified the sentence to avoid confusion. Please, see the new sentence in the Discussion section of the new version submitted.

However, after the introduction of the pneumococcal conjugate vaccines (PCV7 and later PCV13), a rise of non-vaccine serotypes displaying antibiotic-resistant has been identified (21,22). A clear example of this phenomenon is represented by serotype 19A, (included in PCV13 but not in PCV7) that was one of the most prevalent PENR serotypes after the use of PCV7 (23).”

Line 172. Replace “descript” with “described” o similar.

Ok. Done

Line 179. “…because divert very efficiently the host immune response (30).” The reference mentioned does not exactly say that; they say in their abstract that “A different pattern of evasion of complement immunity and phagocytosis was observed between (serotype 11A) genotypes.”. As in the present work there is not study of genotypes within serotype 11A, this sentence should be better explained.

We agree with the Reviewer that the paragraph should be better explained indicating the relevance of the genotype ST652111A that has spread all across Europe, is one of the most frequent and it has an increased ability to avoid the host immune response. Changes made in Discussion:   

Hence, the emergence of penicillin-resistant strains of serotype 11A is concerning from a pathogenesis perspective (31). The invasive disease potential by this serotype is highly related to the rise of genotype ST652111A that has spread all across Europe in the last years becoming one of the most prevalent within serotype 11A (Gonzalez Diaz A, Euro Surveill. 2020 Apr;25(16):1900457. doi: 10.2807/1560-7917.ES.2020.25.16.1900457), This genotype of serotype 11A is associated to high levels of antibiotic resistant, shows a greater ability to form biofilms and avoids very efficiently the host immune response (31).

Line 191. “…can help to prevent the most difficult cases of IPD”.  Please, rewrite clearly the meaning of “most difficult cases”: most severe, most antibiotic-resistant, …

The paragraph (lines 189-192) has been changed:

The surveillance of the behaviour of serotypes and their resistance to antibiotics will be crucial for the development of new vaccines and the implementation of vaccination schedules that can help to prevent IPD.

Line 192. “The use of vaccines in national immunization schedules is a cost-effective measure to decrease antibiotic resistance”. Please, add a reference that demonstrates this assessment.

The sentence has been modified:

The use of vaccines in national immunization schedules is a cost-effective measure to decrease antibiotic resistance (Andrejko K, Lancet Microbe. 2021 Sep;2(9):e450-e460. doi: 10.1016/S2666-5247(21)00064-1; Atkins KE, Flasche S. Lancet Glob Health 2018; 6: e252.) and probably, the inclusion of serotypes 11A and 24F in future vaccines of broader spectrum of coverage could be crucial for preventing penicillin resistance and non-susceptibility (36,37).

Lines 212-217. “To analyze the evolution of serotypes causing IPD and the pattern of penicillin susceptibility during the period 2007 to 2021, the Odds Ratio (OR) with its correspondent 95% confidence intervals (CI95) were calculated.”

It is not clear to this reviewer the need of the OR calculation in tables 1 and 2. What do they represent? In general, the odds ratio is used to measure an association between an exposure and an outcome (case-control studies). Other statists could be more appropriate for the comparison of frequencies.

The odds ratio and its confidence interval was calculated to evaluate the association between each serotype and the presence or absence of antibiotic resistance

Lines 214-216. “The extended Mantel-Haenszel Chi Square for Trend (Chi Trend) was used to evaluate the evolution along time of serotypes in comparison to the rest of the penicillin resistant or PNSSDR strains.”

Please, replace “penicillin resistant” with the PENR abbreviator used all through the text.

Ok. Done

I cannot see any analysis of the evolution along time of serotype in the text. If no results of this analysis are going to be shown, this can be deleted from Material and methods.

Deleted from material and methods

Reviewer 2 Report

Antibiotics-Article revision: “Evolution of the antimicrobial susceptibility to penicillin in invasive strains of Streptococcus pneumoniae during 2007-2021 in Madrid, Spain”

The aim of the study was to analyze the evolution of the phenotypical profiles of antimicrobial susceptibility to penicillin in all Invasive Pneumococcal Disease strains isolated in Madrid, Spain, during 2007-2021.

General observations

This study describes the serotype distribution of S. pneumoniae invasive isolates from 2007 to 2021 in Madrid. A total of 7,133 isolates were studied, revealing 2% of penicillin highly resistant isolates and 24% of non-susceptible at standard dosing regimen isolates. In addition, 94.4% of all the PENR belonged to four serotypes including 11A (33.6%), 19A (30.8%), 14 (20.3%) and 9V (9.8%). All the strains of serotype 11A which is a non-PCV13 serotype were detected after the year 2011. The changes in S. pneumoniae serotypes distribution reflect the use of conjugate vaccines, causing the emergence of non-PCV13 pneumococcal strains with different PENR or PNSSDR patterns. In particular, the emergence of serotype 11A resistant to penicillin is worrisome.

I have some minor points:

- The use of Etest for penicillin susceptibility in S. pneumoniae represents a EUCAST warning. This should be specified as a limitation of the study.

- Figure 1: the introduction of a specific vaccine should be indicated in the figure in correspondence of the year (also with arrows).

- A Figure or a Table with vaccine typology and serotype vaccine coverage should be added.

- Susceptibility to ceftriaxone (or cefotaxime) should be provided for isolates not-susceptible to penicillin.

- Facultative: also data susceptibility to levofloxacin and eritromycin for penicillin not-susceptible isolates could be interesting to describe.

Author Response

Dear Reviewer,

It is a pleasure for us to have the opportunity to present our results in this journal. We appreciate and we receive all suggestions with interest to improve our work. We hope that all the answers and changes made are to your liking and we hope that this work will be interesting for publication in this journal.

We present below the anwers to each point but we hace also attached all the answers in the attached document in case it is easier for you to review.

REVIEW:

I have some minor points:

- The use of E test for penicillin susceptibility in S. pneumoniae represents a EUCAST warning. This should be specified as a limitation of the study.

In the discussion, a new sentence has been introduced:

Although E-test has been widely used to perform antimicrobial susceptibility testing of S. pneumoniae, this method is not considered by CLSI or EUCAST in their guidelines, and therefore, it may represent a limitation for the curent study

 - Figure 1: the introduction of a specific vaccine should be indicated in the figure in correspondence of the year (also with arrows).

In the new version, we have indicated the introduction of pneumococcal vaccines using arrows

- A Figure or a Table with vaccine typology and serotype vaccine coverage should be added.

We appreciate the suggestion of the reviewer. The serotype vaccine coverage was fully descripted in the introduction section.

- Susceptibility to ceftriaxone (or cefotaxime) should be provided for isolates not-susceptible to penicillin.

We respond next to the following observation

- Facultative: also, data susceptibility to levofloxacin and eritromycin for penicillin not-susceptible isolates could be interesting to describe.

We agreed with this reviewer comment, and the susceptibility to cefotaxime is included in our systematic work routine. However, these data were out of the aim of this study and were not included in the text. The same appreciation is applicable to other antibiotics such as levofloxacin or erythromycin.

Reviewer 3 Report

The authors analyzed the evolution of the phenotypical profiles of antimicrobial susceptibility to penicillin (PEN) in all IPD strains isolated in Madrid, Spain, during 2007-2021. A total number of 7,133 invasive clinical isolates were characterized between 2007 and 2021. Ac-cording to CLSI and EUCAST breakpoints the strains were categorized as PEN resistant (PENR) when the minimum inhibitory concentration (MIC) was >2mg/L or penicillin non-susceptible at standard dosing regimen (PNSSDR) when their MIC was > 0.06 mg/L. Levels of PENR and PNSSDR were 2.0% and 24.2% respectively. In addition, 94.4% of all the PENR belonged to four serotypes including 11A (33.6%), 19A (30.8%), 14 (20.3%) and 9V (9.8%). All the strains of sero-type 11A which is a non-PCV13 serotype were detected after the year 2011. Serotypes 6C, 15A, 23B, 24F, 35B, 19F, 16F, 6B, 23F, 24B, 24A, 15F and a limited number of strains of serogroups 16 and 24 (non-typed at serotype level) were associated to PNSSDR (p<0.05). PNSSDR strains of non-PCV13 serotypes 11A, 24F, 23B, 24B, 23A and 16F were more frequent from 2014 to 2021.

The changes in S. pneumoniae serotypes distribution associated to the use of conjugate vaccines had caused in our region the emergence of non-PCV13 pneumococcal strains with different PENR or PNSSDR patterns. The emergence of serotype 11A resistant to penicillin as the most important non-PCV13 serotype is a worrisome event with marked relevance from the clinical and epidemio-logical perspective.

The paper will be ready for publication after major revision.

Please highlight your contributions in introduction.

I think the introduction should be supported by the application of machine learning and artificial intelligence in healthcare field you may check:

Artificial intelligence for forecasting the prevalence of COVID-19 pandemic: an overview

Antibiotic discovery with machine learning

All these publications may support the introductions section.

The paper is well-written, I have to thank you to your effort.

What are the main features in Figure 1?

The abstract should be rewritten to reflect the significance of the proposed work. The current abstract shows a lot of background information.

Conclusion: What are the advantages and disadvantages of this study compared to the existing studies in this area?

The inspiration of your work must further be highlighted. Some suggested recent literatures should add.

Add future works as bullets.

“The variation in the incidence and antibiotic resistance of this serotype has been very evident with the introduction of each vaccine in the childhood immunization schedule (3,22,23).”; why?

“In last years, there has been a decrease in the incidence of many of the resistant strains but also in the incidence of IPD due to the COVID-19 pandemic”, add a reference.

The space between value and units may be eliminated.

Looking and wishes for the revised version.

Author Response

Dear Reviewer,

It is a pleasure for us to have the opportunity to present our results in this journal. We appreciate and we receive all suggestions with interest to improve our work. We hope that all the answers and changes made are to your liking and we hope that this work will be interesting for publication in this journal.

We present below the anwers to each point but we hace also attached all the answers in the attached document in case it is easier for you to review.

REVIEW:

The paper will be ready for publication after major revision.

Please highlight your contributions in introduction.

The introduction contained up to 8 different references. Among these publications, references 1, 3, 4, 5 and 7 were contributions from some of the authors of the current manuscript. In the new version, we have added one additional contribution published very recently in this special issue (Redondo, E et al. Antibiotics 2023, 12, 138. https://doi.org/10.3390/antibiotics12010138)

I think the introduction should be supported by the application of machine learning and artificial intelligence in healthcare field you may check:

Artificial intelligence for forecasting the prevalence of COVID-19 pandemic: an overview

Antibiotic discovery with machine learning

All these publications may support the introductions section.

At the end of the introduction of the new version submitted we have included these suggestions.

Antimicrobial resistance has been proposed as one of the top ten public health threats by the World Health Organization. To reinforce the discovery of new antibiotics, artificial intelligence strategies have been proposed to identify new antibiotics and even predict the evolution of vaccine preventable diseases (de la Fuente-Nunez, C. Antibiotic discovery with machine learning. Nat Biotechnol 40, 833–834 (2022). https://doi.org/10.1038/s41587-022-01327-w; Ma Y, et al. Nat Biotechnol. 2022 Jun;40(6):921-931. doi: 10.1038/s41587-022-01226-0; Elsheikh, A.H.; Saba, A.I.; Panchal, H.; Shanmugan, S.; Alsaleh, N.A.; Ahmadein, M. Artificial Intelligence for Forecasting the Prevalence of COVID-19 Pandemic: An Overview. Healthcare 2021, 9, 1614. https://doi.org/10.3390/healthcare9121614).

In this context of high rates of antibiotic resistance and serotype replacement by non-vaccine serotypes, the aim of this study was to analyze the evolution of the phenotypical profiles of antimicrobial susceptibility to penicillin in IPD isolates from Madrid, Spain, during the period 2007-2021.

The paper is well-written, I have to thank you to your effort.

Thank you.

What are the main features in Figure 1?

The main findings of figure1 are explained after table 3 in the results section. In this paragraph we explain.

We have added references in the results section that are supported by Figure 1.

The abstract should be rewritten to reflect the significance of the proposed work. The current abstract shows a lot of background information.

The abstract contains the major findings of the current manuscript. Following the Reviewer’s suggestion we have deleted the background information in the Abstract section related to the breakpoints considered by CLSI and EUCAST because this paragraph is background information that is already explained in the methods section and we have added a new table (requested by Reviewer 1) indicating the breakpoints.

Conclusion: What are the advantages and disadvantages of this study compared to the existing studies in this area?

In different parts of the discussion section, we emphasize some of the findings of the current manuscript in comparison with other studies even at national level. In this sense, we discuss the importance of serotype 19A from our study and we discuss our results with other studies. Please see lines 156-160.

Another example is the importance of serotype 11A where we discuss our results in the context of other studies (lines 174-180).

In addition, we also explain the importance of serotype 24F (lines 184-186) and even the impact of COVID-19 pandemic in antibiotic resistance (lines 188-195).

The inspiration of your work must further be highlighted. Some suggested recent literatures should add.

In the discussion section we have mentioned and discussed several findings described by several authors of this manuscript (original references 20, 22, 30 and 31). In the new version, we have added a recent study published in this special issue (Redondo, E.; Rivero-Calle, I.; Mascarós, E.; Ocaña, D.; Jimeno, I.; Gil, Á.; Díaz-Maroto, J.L.; Linares, M.; Onieva-García, M.Á.; González-Romo, F.; Yuste, J.; Martinón-Torres, F. Vaccination against Community-Acquired Pneumonia in Spanish Adults: Practical Recommendations by the NeumoExperts Prevention Group. Antibiotics 2023, 12, 138. https://doi.org/10.3390/antibiotics12010138).

“The variation in the incidence and antibiotic resistance of this serotype has been very evident with the introduction of each vaccine in the childhood immunization schedule (3,22,23).”; why?

The introduction of PCV7 caused an increase of serotype 19A whereas the introduction of PCV13 (that covered serotype 19A) decreased the incidence of this serotype and the number of strains of serotype 19A harboring antibiotic resistance.

“In last years, there has been a decrease in the incidence of many of the resistant strains but also in the incidence of IPD due to the COVID-19 pandemic”, add a reference.

Ok. Done

The space between value and units may be eliminated.

Ok. Done

Looking and wishes for the revised version.

Round 2

Reviewer 1 Report

The authors have adequetely addressed all the questions raised by this reviewer

Reviewer 3 Report

Accept in present form.